# Stokes flow analogous to viscous electron current in graphene

Jonathan Mayzel[1], Victor Steinberg [1,2] & Atul Varshney [1,3]

Electron transport in two-dimensional conducting materials such as graphene, with dominant electron–electron interaction, exhibits unusual vortex flow that leads to a nonlocal current-field relation (negative resistance), distinct from the classical Ohm's law. The transport behavior of these materials is best described by low Reynolds number hydrodynamics, where the constitutive pressure–speed relation is Stoke's law. Here we report evidence of such vortices observed in a viscous flow of Newtonian fluid in a microfluidic device consisting of a rectangular cavity—analogous to the electronic system. We extend our experimental observations to elliptic cavities of different eccentricities, and validate them by numerically solving bi-harmonic equation obtained for the viscous flow with no-slip boundary conditions. We verify the existence of a  predicted threshold at which vortices appear. Strikingly, we find that a two-dimensional theoretical model captures the essential features of three-dimensional Stokes flow in experiments.

[1] Department of Physics of Complex Systems, Weizmann Institute of Science, 76100 Rehovot, Israel. [2] The Racah Institute of Physics, Hebrew University of Jerusalem, 91904 Jerusalem, Israel. [3] Institute of Science and Technology Austria, Am Campus 1, 3400 Klosterneuburg, Austria. Correspondence and requests for materials should be addressed to A.V. (email: atul.varshney@ist.ac.at)

Electron transport in conducting materials is often hindered by lattice disorder, impurities, and interactions with phonons and electrons[1]. In ultraclean two-dimensional (2D) materials, the transport is primarily affected by electron–phonon ($\gamma_p$) and electron–electron ($\gamma_{ee}$) scattering processes; $\gamma$ is the scattering rate of each process. In the limit $\gamma_p \gg \gamma_{ee}$, a fast momentum relaxation of electrons results in a linear relationship between local current and applied electric field called Ohm's law. This relationship breaks down when the momentum exchange rate of electrons with each other is much faster than with the lattice[2] ($\gamma_{ee} \gg \gamma_p$). In this regime, strongly interacting electrons move in a neatly coordinated manner which resembles the flow of viscous fluids[3,4]. Thus, hydrodynamic equations can be employed to describe the transport behavior of electron flow provided mean free path ($\ell_{ee}$) for momentum-conserving electron–electron collisions is the shortest length scale in the problem, i.e., $\ell_{ee} \ll w, \ell_p$; where $w$ is the system size and $\ell_p$ is the mean free path for momentum-nonconserving electron–phonon collisions[3–7]. The first experimental signature of hydrodynamic electron flow was obtained in the measurements of differential resistance of electrostatically defined wires in a 2D electron gas in (Al,Ga)As heterostructures[8,9].

Recent development in the synthesis of ultrapure crystals has facilitated the investigation of the viscous flow regime of electrons at elevated temperatures, so called viscous electronics[10–13]. Experiments on viscous electron flow through narrow constrictions in doped single- and bi-layer graphene reveal anomalous (negative) local resistance[10]. It was understood that the negative resistance may arise due to viscous shear flow which generates a vortex (whirlpool) and a backflow producing a reverse electric field that acts against the applied field driving the source-drain current[2,10,14,15]. Concurrently, Levitov and Falkovich (L&F) developed a theoretical model based on Stokes flow of strongly interacting electrons[2,16]. They explored three transport regimes of electron fluids, namely: Ohmic, mixed ohmic-viscous and viscous, in an infinitely long 2D rectangular strip, with point source and drain contacts located at the center, on opposite sides of the strip. Linearized 2D Navier-Stokes equation in the limit of low Reynolds number (Re ≪ 1) is used to describe the transport regimes of incompressible electron fluids that yields bi-harmonic stream function, in contrast to the harmonic stream function for the Ohmic case. Here, the dimensionless Reynolds number (Re) defines the ratio of inertial and viscous stresses in fluids[4]. Furthermore, they extended their model for viscous electron flow in a three-dimensional (3D) conducting slab of small, finite thickness, and showed that the extra dimension in the Stokes equation translates into an effective resistance term[2].

A steady, viscous, and incompressible Newtonian fluid flow in a 3D slab of width $w$ and height $h$ at Re ≪ 1 is described as $\eta(\partial_x^2 + \partial_y^2 + \partial_z^2)u_i = \nabla P$, where $u_i(x,y,z) = u_i(x,y)u_z(z)$, $\eta$ is the dynamic viscosity of the fluid and $P$ is the pressure field[3]. As a vertical parabolic velocity profile $u_z = z(h-z)/h^2$ is globally uniform, and $\nabla P(z) = \text{const}$ due to the uniformity of the velocity in both spanwise ($x$) and streamwise ($y$) directions, one gets similar to what is suggested in ref. [2] the following 2D linearized stationary Navier-Stokes equation after integration of the vertical velocity profile: $[\eta(\partial_x^2 + \partial_y^2) - 12\eta/h^2]u_i(x,y) = 6\nabla P$. The threshold value to realize a vortex in a rectangular slab of thickness $h$ is $\varepsilon \equiv 12w^2/h^2 \leq 120$ and it arises when the viscous shear force exceeds the wall friction (Ohmic) due to the boundaries. Therefore, in a 3D system the viscous effect will be more pronounced when the system is thicker, at the same width, since the friction arising from the top and bottom walls will be less significant. The criterion value to observe vortices, i.e. $\varepsilon < \varepsilon_c$, is estimated numerically for the rectangular slab in ref. [2] (see also Supp. Info. therein).

The generation of vortices in a fluid flow is typically associated with high-Re flow and inertial effects. However, such inferences are likely based on the incorrect notion that low-Re flow is irrotational, which is only applicable to an ideal fluid without viscosity, where the Kelvin circulation theorem is valid[3,17]. Thus, it is evident that a non-potential or rotational flow bears vorticity, but to produce vortices in Stokes flow requires substantial efforts due to a strong dissipation of vorticity at Re ≪ 1. Strikingly, in a wall-dominated microfluidic channel flow, vortices could be generated at low aspect ratio, $w/h$, of the channel despite a significant wall friction and viscous dissipation[2].

Here, motivated by the observation of a distinct vortex flow in a strongly interacting electron system discussed above, we perform experiments on a viscous flow of ordinary Newtonian fluid, at low-Re, in a microfluidic device consisting of a rectangular cavity, analogous to the 2D electronic system. Indeed, we observe a pair of symmetric vortices in the cavity region, when the geometrical criterion for the vortex observation $\varepsilon < \varepsilon_c$ is satisfied, in agreement with the predictions[2] of L&F. Further, we expand our observations to elliptical cavities of different eccentricities ($e$) and verify them with the analytical predictions[2] by numerically solving the bi-harmonic equation obtained for the viscous flow with no-slip boundary conditions.

## Results

**Rectangular cavity.** A long-exposure particle streak image in Fig. 1b (see also corresponding Supplementary Movie 1)

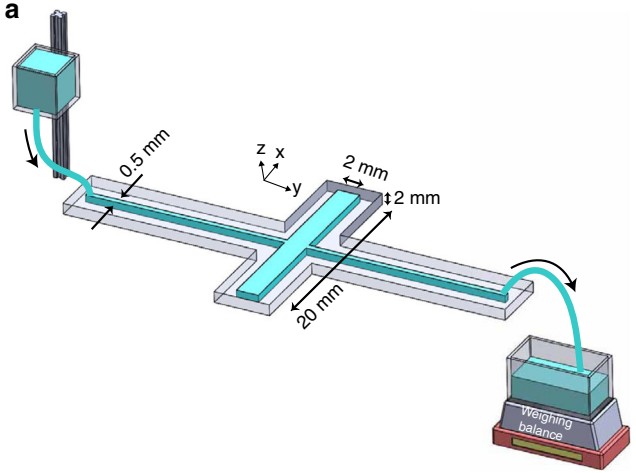

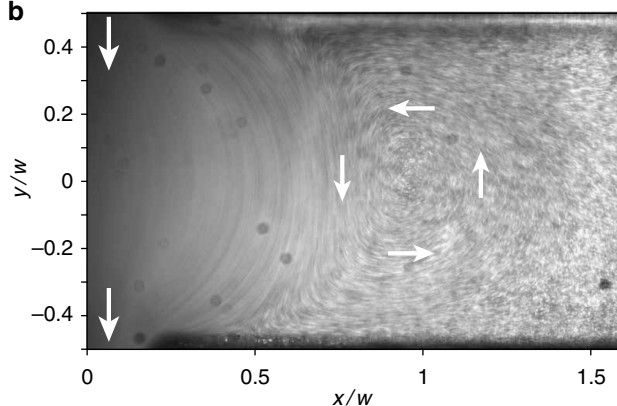

**Fig. 1** Experimental setup and vortex flow in a rectangular cavity. **a** Schematic of the experimental setup (not to scale). **b** Particle streaks of the flow in a rectangular cavity ($e \to 1$) at Re = 0.07; see also corresponding Supplementary Movie 1. White arrows indicate the flow direction

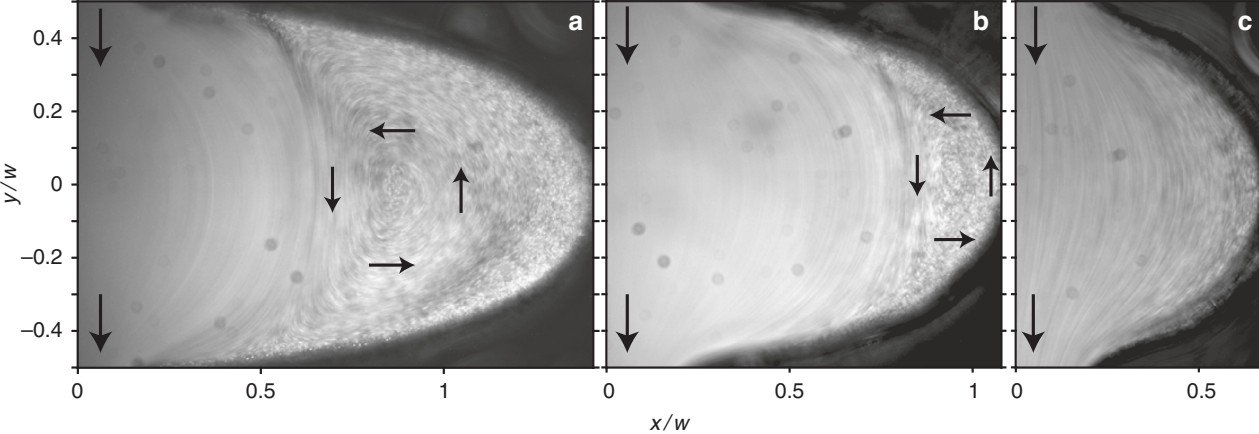

**Fig. 2** Long-exposure particle streaks of the flow in cavities of different eccentricity values. Snapshots of the flow in cavities of **a** $e = 0.95$, Re = 0.08; **b** $e = 0.9$, Re = 0.07; and **c** $e = 0.75$, Re = 0.02; see also corresponding Supplementary Movies 2–4. Black arrows indicate the flow direction

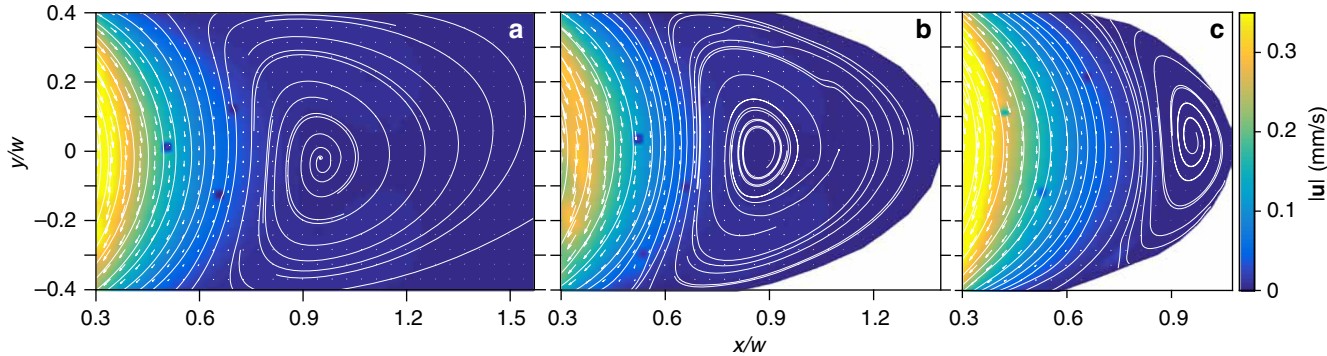

**Fig. 3** Velocity field obtained from micro-particle image velocimetry for different eccentricities. Color map of fluid velocity magnitude $|\mathbf{u}|$ obtained by $\mu$PIV in the cavity region for **a** $e \rightarrow 1$, Re = 0.07; **b** $e = 0.95$, Re = 0.08; and **c** $e = 0.9$, Re = 0.07. Velocity field and streamlines are shown by arrows and lines, respectively

illustrates the vortex flow at Re = 0.07 in the rectangular cavity region ($e \rightarrow 1$) of the device shown in Fig. 1a and described in Methods section. A pair of symmetric vortices appears in the cavity regions; here we show only one side of the cavity. The flow in elliptic cavities is shown in Fig. 2 and discussed in the next section. For the rectangular cavity, the streamlines and velocity field, obtained from micro-particle image velocimetry ($\mu$PIV), of the vortex flow (Fig. 3a) resemble the structures of the current streamlines and potential map as analyzed by L&F for viscous flow in an infinite long 2D rectangular strip[16]. Furthermore, we obtain the location of the vortex center at $x_0 \approx 0.96w$ (see Figs. 1b and 3a), which is in fair agreement with the analytical prediction[16] of $x_0 \approx w$.

**Elliptic cavities.** Next, we investigated elliptical cavities with different eccentricities, $e = \sqrt{1 - b^2/a^2}$, where $a$ and $b$ are the major and minor axes of an ellipse, respectively. Figure 2a, b show long-exposure images depicting the vortex flow for $e = 0.95$ (at Re = 0.08) and $e = 0.9$ (at Re = 0.07), respectively; see also Supplementary Movies 2–4. The corresponding streamlines and velocity field are shown in Fig. 3b, c. For $e = 0.95$, the vortex center appears at $x_0 \approx 0.85w$ (see Figs. 2a and 3b), and for $e = 0.9$ the vortex is pushed towards the cavity edge and its center is located at $x_0 \approx 0.95w$ (see Figs. 2b and 3c). However, for $e = 0.75$ we do not observe a vortex (Fig. 2c and Supplementary Movie 4), which is discussed further.

**Analytic solution and numerical simulations.** Using the Stokes equation, the incompressibility condition, and the fact that the system is 2D, we introduce the streamline function $\psi(x,y)$ via 2D velocity $\mathbf{u}(x,y) = \mathbf{z} \times \nabla \psi$, which reduces the Stokes equation to the bi-harmonic equation[18,19] $(\partial_x^2 + \partial_y^2)^2 \psi(x,y) = 0$. We solve the bi-harmonic equation analytically for disk geometry and numerically for elliptical geometries with no-slip boundary conditions $u_y = u_x = 0$, where $u_y$ and $u_x$ are transverse and longitudinal components of $\mathbf{u}$, respectively. For a disk geometry ($e = 0$) of radius $R$ centered at the origin and the flow is injected (collected) with speed $u_0$ at $y = R$ ($y = -R$), we do not observe vortices in the flow field (Supplementary Figure 1a,b). However, the real part of the pressure field

$$P(x,y) = \eta u_0 \Re \left( \frac{8iR^2}{\pi} \frac{\bar{z}}{(\bar{z}^2 + R^2)^2} \right), \qquad (1)$$

where $\bar{z} = x - iy$, exhibits non-trivial patterns (Supplementary Figure 1c).

Further, we consider an ellipse with major axis $a$ and minor axis $b$, where flow is injected and collected with a speed $u_o$ at $(0, \pm b)$. Since vortices did not appear for a disk ($e = 0$), and did appear for an infinite rectangular strip ($e \rightarrow 1$) we expect that at some critical value of eccentricity vortices would start to emerge. The streamline function for the ellipse is obtained using a multi-step computational approach: First we define the vorticity of the flow

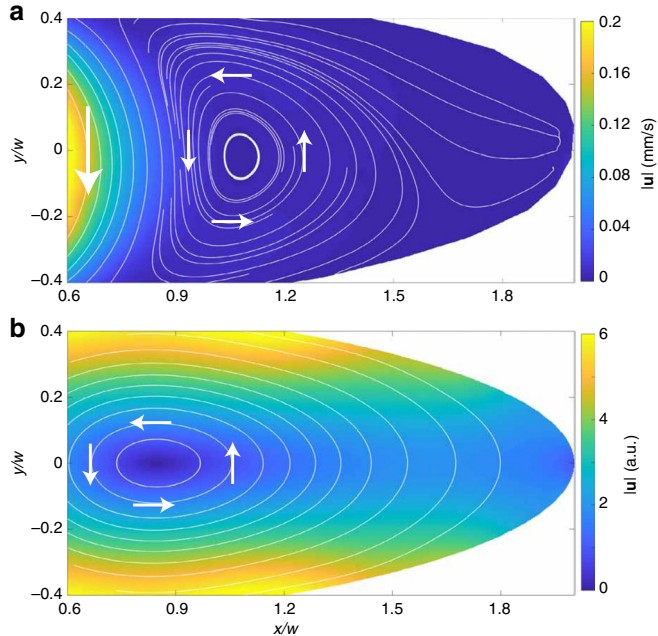

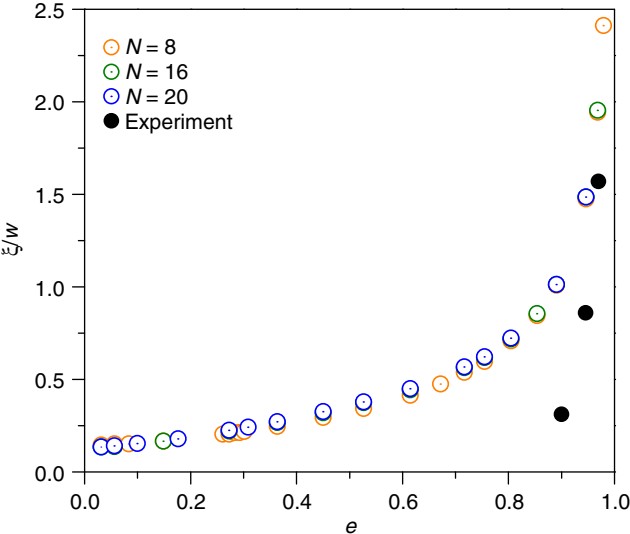

**Fig. 4** Experimental and numerical results for $e = 0.97$. Color map of fluid velocity magnitude $|\mathbf{u}|$ in the cavity region for $e = 0.97$ from **a** experiment and **b** numeric with $N = 16$. Velocity streamlines and flow direction are shown by lines and arrows, respectively

**Fig. 5** Separatrix distance vs. $e$. Normalized vortex separatrix distance from the cavity edge, $\xi/w$, as a function of eccentricity ($e$), obtained from experiments (solid symbols) and from numerics by solving bi-harmonic stream function of order $N = 8$, 16, and 20 (open symbols)

field $\omega = \nabla \times \mathbf{u}$, which is used to obtain the complex harmonic function

$$\Phi(\bar{z}) = \frac{1}{\eta}P + i\omega. \tag{2}$$

Since $\frac{1}{\eta}P$ and $\omega$ form a Cauchy–Riemann pair, we can use conformal mapping to map $\Phi$ from a disk to an ellipse. The mapping formula is given by[20,21]:

$$\kappa = \sqrt[4]{m}\, \mathrm{sn}\left(\frac{2K(m)}{\pi}\sin^{-1}\left(\frac{z}{\sqrt{a^2 - 1}}\right), m\right), \tag{3}$$

where sn is the Jacobian elliptic sine, and $K(m)$ is the complete elliptic integral of the first kind with modulus $m$. This formula maps the interior of an ellipse in the $z$-complex plane to the interior of a disk in the $\kappa$-complex plane. Then, from $\Phi$ in the elliptical geometry, the real part of the vorticity is extracted as $\omega = \Re(\Phi)$ and then an inverse Laplacian is applied on $\omega$ that yields $\psi$ up to a harmonic function $\psi = \psi_0 + F$, where $\psi_0$ is the result of the inverse Laplacian and $F$ is a general harmonic function determined by boundary conditions. The most general form of a harmonic function in elliptic coordinates $(\mu, \nu)$ that preserves the symmetries in the problem is given by:

$$F(\mu, \nu) = \sum_{n=0}^{N}(A_n e^{-n\mu} + B_n e^{n\mu})\cos n\nu. \tag{4}$$

The coefficients $A_n$ and $B_n$ are determined by fitting $\psi$ to the no-slip boundary conditions, and $N$ is the highest order for which the coefficients are computed. A detailed numerical scheme is delineated in Supplementary Notes. Results from numerics show that vortices exist for any ellipse with non-zero eccentricity, for instance, see Fig. 4b for $e = 0.97$ and corresponding experimental results in Fig. 4a. In addition, we show velocity fields computed numerically and obtained experimentally for $e = 0.9$ and 0.95 in Supplementary Figures 2 and 3, respectively.

Next, we compute the distance of separatrix from the edge of the ellipse normalized with the cavity width, i.e. $\xi/w$, for different elliptical cavities. Figure 5 shows the variation of $\xi/w$ as a function of $e$, ranging from a disk ($e = 0$) to an infinite rectangular strip ($e \to 1$), obtained numerically from the fitted harmonic function of different orders $N$ (open symbols). Based on Fig. 5, it suggests that, for a disk, a vortex "lies" on the disk edge, and as the disk is deformed into an elliptical geometry the vortex escapes the edge and develops inside the ellipse. The experimental results in Fig. 5 (solid symbols) exhibit the same trend as from the numerical simulations. There is though a significant quantitative discrepancy between the experimentally measured $\xi/w$ and the computed value.

## Discussion
The quantitative discrepancy in the $\xi/w$ dependence on $e$ can likely be explained by the different flow regimes considered in the experiment and numerical simulations. The numerical simulations were conducted for the 2D viscous flow without any wall friction and with zero size inlet-outlet channels (in analogy with the electrical transport corresponding to a pure viscous case with a zero contact size), whereas the experimental flow is 3D with finite-size inlet-outlet channels (0.5 mm channel width compared to the cavity width of ~2 mm). The latter flow can be approximated by a quasi-2D viscous flow with the friction term $\varepsilon = 12w^2/h^2$, corresponding to a mixed viscous-ohmic case.

As discussed by L&F[2], in the mixed viscous-ohmic regime, vortices will form when the resistance to viscosity ratio $\varepsilon$ is smaller than some critical value $\varepsilon_c$, which is geometry dependent. In our experimental system, $\varepsilon < \varepsilon_c$ for the rectangular cavity with $\varepsilon = 12$ (see Methods section) and for elliptic cavities, the value of $\varepsilon_c$ is not known and probably further depends on the eccentricity $e$; $\varepsilon < \varepsilon_c$ holds true only for $e \to 1$. Thus, the experimental results for the rectangular strip $\varepsilon/\varepsilon_c = 0.1 \ll 1$ match quite well with the analytical predictions (Fig. 3a and related text). For ellipses with varying eccentricity $e$, this critical threshold is probably lower and reduces further with decreasing $e$. For $e > 0.9$, $\varepsilon_c$ is possibly still higher than 12, so we do observe vortices, even though their size and streamlines do not match with the numerical solution of the purely viscous case. A comparison between the numerical and

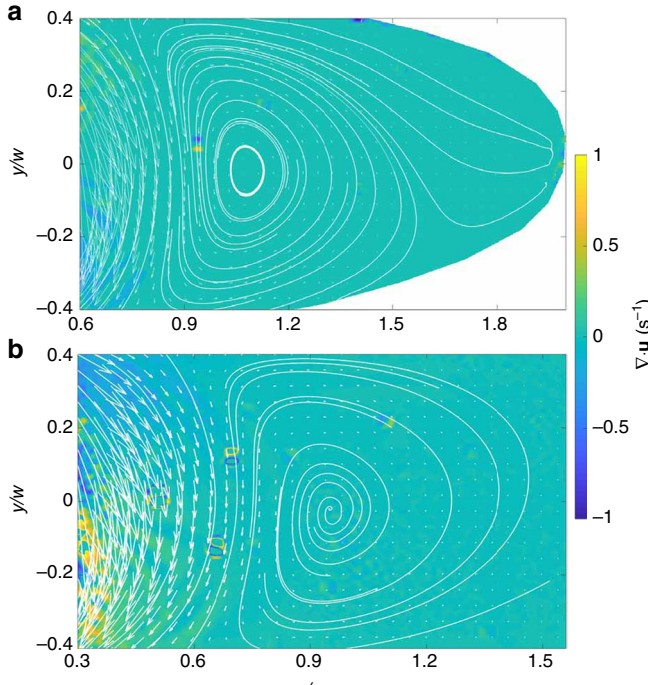

**Fig. 6** Divergence of the velocity field obtained from experiments. Divergence field (in color) obtained from measured velocity field (shown by arrows) for **a** elliptical cavity of $e = 0.97$ and **b** rectangular cavity. Streamlines are shown by the white lines. Other velocity fields ($e = 0.9$ and $0.95$) also show zero average divergence (see Supplementary Figure 5).

experimental results of the velocity magnitude and flow field for ellipse with $e = 0.97$ can be seen in Fig. 4. For ellipse with $e \sim 0.75$, the critical threshold is probably $< 12$, therefore we do not observe vortices. This issue requires further investigation by solving the equation for the mixed viscous-ohmic case for an ellipse with no-slip boundary conditions and by conducting experiments for a fixed value of the eccentricity $e$ and various values of $\varepsilon$.

By using the condition for the vortex generation, which is derived in the quasi-2D approximation of a 3D channel flow, we assume that the 2D model is applicable to characterize the 3D flow. The results of vortex observation in 3D flow indicate the validity of this assumption and its applicability. To further validate this assumption we compute the divergence of the 2D velocity field close to the mid-height of the cavity ($\nabla \cdot \mathbf{u}$) based on the experimental data. As can be seen in Fig. 6, the divergence field is close to zero in the cavity regions which suggests a 2D nature of the flow. The non-zero deviations in the divergence field are produced either by velocity fluctuations since they appear in plus/minus pairs or by the lower accuracy of PIV measurements in the region close to the inlet and outlet, where the velocities are high. A comparison between the divergence fields computed from experimentally and numerically obtained velocity fields for $e = 0.97$ is shown in Supplementary Figure 4.

In conclusion, we have numerically examined the 2D flow of a Newtonian fluid by using an experimental, 3D flow at $Re \ll 1$; the latter can be reduced to a quasi-2D flow with the wall friction term, as suggested in ref. 2. We have shown that vortices do appear in a rectangular cavity when the criterion $\varepsilon < \varepsilon_c$ is satisfied, verifying the analytical prediction of L&F with a good accuracy. Furthermore, we expand the analytical predictions to disk and elliptic geometries, and validate the latter by experimental observations of vortices and the normalized vortex separatrix distance $\xi/w$. The observed quantitative discrepancies in $\xi/w$ as a

function of the eccentricity may be explained by a reduction of the criterion value for the onset of the vortex observation $\varepsilon_c$ with decreasing eccentricity, $e$, from unity for a rectangular cavity down to zero for a circular disk. The criterion $\varepsilon < \varepsilon_c$ of obtaining a vortex breaks down at some value of $e$, which leads to a disagreement with the theory. Thus, conducting experiments in low-Re Newtonian fluid flow allows us to inspect unusual vortex flow properties which are experimentally unobservable in graphene or in other 2D electronic systems. It is important to notice that in a Newtonian fluid flow the boundary conditions are known, simple, and verified, whereas in graphene they are less clear and may evolve due to edge currents, partial slip[22–25], etc.

## Methods

**Experiments.** The experimental system consists of a straight channel ($40 \times 0.5 \times 2$ mm$^3$) endowed with a long rectangular cavity ($e \rightarrow 1$) at the center of the channel, as illustrated in Fig. 1a. The devices are prepared from transparent acrylic glass [poly(methyl methacrylate)]. The width and thickness of the cavity are $w = h = 2$ mm, which gives $\varepsilon = 12$. An aqueous glycerol solution (60% by weight) of viscosity $\eta = 11$ mPa·s at 20 °C is used as a working fluid in the experiments. A smooth gravity-driven flow of the fluid is injected via the inlet into the straight channel, and its flow rate is varied by changing the fluid column height (see Fig. 1a). The fluid exiting from the outlet of the channel is weighed instantaneously $W(t)$ as a function of time $t$ by a PC-interfaced weighing balance (BA-210S; Sartorius) with a sampling rate of 5 Hz and a resolution of 0.1 mg. The flow speed ($\bar{u}$) is estimated as $\bar{u} = \bar{Q}/(\rho w h)$, where time-averaged fluid discharge rate $\bar{Q} = \overline{\Delta W/\Delta t}$, fluid density $\rho = 1156$ Kg m$^{-3}$. Thus, the Reynolds number is defined as $Re = w \rho \bar{u}/\eta$.

**Imaging system.** The fluid is seeded with 2 μm sized fluorescent particles (G0200; Thermo Scientific) for flow visualization. The cavity region is imaged in the midplane directly via a microscope (Olympus IMT-2), illuminated uniformly with a light-emitting diode (Luxeon Rebel) at 447.5 nm wavelength, and a CCD camera (GX1920; Prosilica) attached to the microscope records $10^4$ images with a spatial resolution of $1936 \times 1456$ pixel and at a rate of 9 fps. We employ micro-particle image velocimetry (μPIV) to obtain the spatially resolved velocity field $\mathbf{u} = (u_y, u_x)$ in the cavity region[26]. An interrogation window of $32 \times 32$ pixel$^2$ ($55 \times 55$ μm$^2$) with 50% overlap is chosen to procure $\mathbf{u}$. Experiments are repeated on cavities of different $e$ values.

## Data availability

The data that support the findings of this study are available from the corresponding authors upon reasonable request.

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

## Acknowledgements

J.M. thanks Gregory Falkovich for introducing the problem and his guidance during the studies. We thank Guy Han for technical support. A.V. acknowledges support from the European Union's Horizon 2020 research and innovation program under the Marie Skłodowska-Curie grant agreement No. 754411. This work was partially supported by the Israel Science Foundation (ISF; grant #882/15) and the Binational USA-Israel Foundation (BSF; grant #2016145).

## Author contributions

J.M. and A.V. designed the experiments. J.M. performed the measurements and together with A.V. and V.S. analyzed the data. J.M. performed the analytical and numerical computations. All authors discussed the results and wrote the manuscript.

## Additional information

**Competing interests:** The authors declare no competing interests.

