## [Peer Review File · Nature Communications]

Reviewer #1 (Remarks to the Author):

The authors report on a Stokes flow analogue of the viscous flow of electrons in graphene. The Stokes flow is accessible to measurements and, thanks to the linearity of the equations, amenable to relatively simple analytic derivations. These advantages of the Stokes flow were demonstrated in other systems, such as phonons in a microfluidic crystal, and here they are elegantly utilized to examine theoretical predictions regarding viscous electron flow (ref. 4), which would be much harder to measure in graphene itself. The authors observe in their apparatus the vortex flow that leads to negative resistance in graphene. They further observe similar phenomena in elliptical cavities and compare them to a simple quasi-2D Stokes flow model. Overall, I find the measurement beautiful, and the analogy to electron transport deep and intriguing. Hence, in principle, I think that the manuscript deserves publication after the authors consider a few points.

The main misgiving is about the discrepancy between the quasi-2D model and the experiment regarding the position of the vertex for varying geometry (esp. figure 5). The authors are very frank about it and give good reasons for the discrepancy, mainly the finite size of the inlet and outlet and the 3D nature of the flow. Perhaps the authors could tweak the geometry to be closer to these simplifying assumptions (such as wider channel). The manuscript could also benefit from serious editing and rebalancing of the discussion to include more intuition at the (for example, as to why there is a critical threshold for the emergence of a vertex). The text is clunky and hasty at times with quite a few typos (e.g., u is missing in p.1 c.2, 5 lines from the end) which are for the authors to correct.

Reviewer #3 (Remarks to the Author):

Review on "Stokes flow analogous to viscous electron current in graphene".

The authors study the flow of viscous fluid through a narrow quasi-2D channel equipped with rectangular and elliptical cavities adjacent to the channel. The manuscript reports the observation (photos and videos) of vortices which emerge as a result of viscous entrainment between the fluid flowing in the channel and that present in the cavities. In addition, the authors provide a theoretical support for their studies by solving numerically and analytically equations of fluid dynamics. The three-dimensional flow of fluid in the present study resembles that of the recently reported behavior of viscous 2D fluid in graphene and is well-described by 2D hydrodynamics.

The manuscript is well-written, presents a beautiful set of photos and videos and is supported by a theoretical investigation. Despite the outstanding presentation of the results and interesting observations, I am concerned about the significance of the results for both hydrodynamic and condensed matter communities and hope that the authors may help me to recognize them and articulate more clearly in the revised manuscript. The ground for my concern is the following. First of

all, the observation of vortices does not require a special experimental setup to be built – one can simply look at the flow of water in a river which may occasionally have some adjacent to the riverbed cavities. It is not a secret, that the vortices, similar to those reported in the manuscript, will naturally form in these cavities. What is it so special that the authors observed in their setup? Second, in my opinion, the comparison with graphene is highly overstated. From the numerous citations of related works on graphene, I concluded that the main message of those works was that electrons, under certain conditions, may behave as conventional highly-viscous fluids and that this fluid-like behavior has been disregarded for many years and never been observed in solid state devices. In this manuscript, the authors sort of close the loop by stating that the classical Newtonian viscous fluid (aqueous glycerol solution) behaves as graphene's electrons, which in turn exhibit classical fluid behavior. This comparison sounds very far-fetched to me and I hope the authors will be able to comment on what is it that makes the flow of aqueous glycerol solution in narrow channels so special and unexpected.

Apart from the major points raised above, there are several questions/comments that remain unclear to me. Here they are:

1. First of all, the authors studied the flow under relatively low Re numbers (<1). Is it possible to detect any signatures of the (pre)turbulent flow by changing the flow velocity and fluid viscosity (e.g. by a local heater placed under the cavity) and going to the regime of large Re in the experimental setup shown in Fig. 1? If probed, these observations may be very useful for both hydrodynamic and graphene communities, especially in the light of several reports predicting turbulence in electronic systems (arXiv:1807.07117, Phys. Rev. Lett. 106, 156601).
2. It seems to me the importance of fact that the three-dimensional flow can be well-described by a 2D theoretical model is exaggerated. In the case of a planar flow and when the slab width is larger than h , this is i) a very natural fact that stems directly from conventional Navier-Stokes equation after integration over the z -axis and accounting for a parabolic velocity profile across this axis (Landau Lifshitz, Fluid Mechanics) and ii) has been already noted and addressed in detail in Ref. 4. Therefore, I suggest the authors to reconsider the discussion on this topic.

To sum up, I cannot recommend the manuscript in the present form to be published in Nature Communications, yet the reported observations may find their readership in a more special journal such as Phys. Rev. Fluids.

Reviewer #1 (Remarks to the Author):

The authors report on a Stokes flow analogue of the viscous flow of electrons in graphene. The Stokes flow is accessible to measurements and, thanks to the linearity of the equations, amenable to relatively simple analytic derivations. These advantages of the Stokes flow were demonstrated in other systems, such as phonons in a microfluidic crystal, and here they are elegantly utilized to examine theoretical predictions regarding viscous electron flow (ref. 4), which would be much harder to measure in graphene itself. The authors observe in their apparatus the vortex flow that leads to negative resistance in graphene. They further observe similar phenomena in elliptical cavities and compare them to a simple quasi-2D Stokes flow model. Overall, I find the measurement beautiful, and the analogy to electron transport deep and intriguing. Hence, in principle, I think that the manuscript deserves publication after the authors consider a few points.

We are grateful to the reviewer for the positive assessment of our work and recommending the manuscript for publication in Nature Communications.

The main misgiving is about the discrepancy between the quasi-2D model and the experiment regarding the position of the vertex for varying geometry (esp. figure 5). The authors are very frank about it and give good reasons for the discrepancy, mainly the finite size of the inlet and outlet and the 3D nature of the flow. Perhaps the authors could tweak the geometry to be closer to these simplifying assumptions (such as wider channel). The manuscript could also benefit from serious editing and rebalancing of the discussion to include more intuition at the (for example, as to why there is a critical threshold for the emergence of a vertex).

Regarding to the quantitative discrepancy between the quasi-2D model and its 3D experimental verification, we completely agree that the 3D experiment cannot be quantitatively equivalent with the 2D model results. Of course, we thought about varying the channel geometry and in this way to reduce the parameter ϵ corresponding to the Ohmic resistance in the case of electrical resistivity. In the case of hydrodynamic, it is the contribution of the third dimension, i.e. thickness, and one cannot reduce it to zero. The problem with the variation of the geometry is that the threshold, which is known only for a rectangular channel, will be simultaneously varied and we do not know how much. Therefore, it will not contribute more into our quantitative comparison with the 2D model. Further, we followed the Referee advice and added more discussion about the emerging threshold for the vortex appearance and improved the presentation.

The following text has been added in the Introduction section of the manuscript:

“The threshold value to realize a vortex in a rectangular slab of thickness h is $\epsilon \equiv \frac{12w^2}{h^2} \leq 120$ and it arises when the viscous shear force exceeds the wall friction (Ohmic) due to the boundaries. Therefore, in three-dimensional system the viscous effect will be more pronounced when the system is thicker at the same width since the wall friction arising from the top and bottom walls will be less significant. The criterion value to observe vortices, i.e. $\epsilon < \epsilon_c$, is estimated numerically for the rectangular slab in Ref. ⁴ (see Fig. 2 and Supp. Info. therein).”

The text is clunky and hasty at times with quite a few typos (e.g., u is missing in p.1 c.2, 5 lines from the end) which are for the authors to correct.

We have corrected the typo in the revised manuscript version. Also, we have smoothed the text (shown with blue in the revised manuscript) for better readability.

Reviewer #3 (Remarks to the Author):

Review on “Stokes flow analogous to viscous electron current in graphene“.

The authors study the flow of viscous fluid through a narrow quasi-2D channel equipped with rectangular and elliptical cavities adjacent to the channel. The manuscript reports the observation (photos and videos) of vortices which emerge as a result of viscous entrainment between the fluid flowing in the channel and that present in the cavities. In addition, the authors provide a theoretical support for their studies by solving numerically and analytically equations of fluid dynamics. The three-dimensional flow of fluid in the present study resembles that of the recently reported behavior of viscous 2D fluid in graphene and is a well-described by 2D hydrodynamics.

The manuscript is well-written, presents a beautiful set of photos and videos and is supported by a theoretical investigation.

We are glad that the Referee found our set of photos and videos supported by theoretical calculations beautiful, and the presentation of the results and observations outstanding.

Despite the outstanding presentation of the results and interesting observations, I am concerned about the significance of the results for both hydrodynamic and condensed matter communities and hope that the authors may help me to recognize them and articulate more clearly in the revised manuscript. The ground for my concern is the following. First of all, the observation of vortices does not require a special experimental setup to be built – one can simply look at the flow of water in a river which may occasionally have some adjacent to the riverbed cavities. It is not a secret, that the vortices, similar to those reported in the manuscript, will naturally form in these cavities. What is it so special that the authors observed in their setup? Second, in my opinion, the comparison with graphene is highly overstated. From the numerous citations of related works on graphene, I concluded that the main message of those works was that electrons, under certain conditions, may behave as conventional highly-viscous fluids and that this fluid-like behavior has been disregarded for many years and never been observed in solid state devices. In this manuscript, the authors sort of close the loop by stating that the classical Newtonian viscous fluid (aqueous glycerol solution) behaves as graphene’s electrons, which in turn exhibit classical fluid behavior. This comparison sounds very far-fetched to me and I hope the authors will be able to comment on what is it that makes the flow of aqueous glycerol solution in narrow channels so special and unexpected.

The main reason to build the experimental setup is of course to quantitatively verify theory presented in Ref. 4 and in their recent papers. The main key difference with the observation suggested by the Referee and the experimental setup is the value of the Reynolds number. In Nature, it is very difficult to observe a creeping (or Stokes) flow at $Re < 1$. Indeed, simple estimates for water with viscosity $\eta_w = 0.001$ Pa.s to get $Re < 1$, one needs velocity < 1 mm/s and the characteristic length < 1 mm that is not trivial to find and definitely not in a river. For this reason, we used 11 times more viscous fluid than η_w , cavity width of 2 mm and velocities in the range of about 0.1 mm/s. Of course, in this case, we deal only with a laminar (steady) flow and generation of vorticity in this case is not so trivial issue. If one takes an arbitrary geometry channel without taking special care about it, one will not find vortices in the Stokes flow. And the key argument is the threshold value of the geometrical factor $\epsilon = 12(w/h)^2$ which for the rectangular channel geometry is about 120. At $\epsilon > 120$ one will not observe

vortices in the Stokes flow. Another question about the similarity between electrical resistance in graphene and Stokes flow of Newtonian fluid. It is not our theory but the theoretical development started by the publication of Ref. 4. Our experiment and theoretical calculation based on the Navier–Stokes equation at $Re < 1$ are just the experimental verification of this suggestion. We, of course, do not discuss the experimental verifications of this theory by numerous experiments in graphene and in other electronic materials. However, due to similarity in theoretical description and basically the same mathematical equations describing these two phenomena, we believe that our experiment substantiate a real situation in graphene and other analogous two-dimensional electronic systems.

Apart from the major points raised above, there are several questions/comments that remain unclear to me. Here they are:

1. First of all, the authors studied the flow under relatively low Re numbers (< 1). Is it possible to detect any signatures of the (pre)turbulent flow by changing the flow velocity and fluid viscosity (e.g. by a local heater placed under the cavity) and going to the regime of large Re in the experimental setup shown in Fig. 1? If probed, these observations may be very useful for both hydrodynamic and graphene communities, especially in the light of several reports predicting turbulence in electronic systems (arXiv:1807.07117, Phys. Rev. Lett. 106, 156601).

As we described above, our system is designed to conduct experiments at $Re < 1$ and a specially designed channel geometry to quantitatively test the theory: very narrow inlet/outlet, various channel geometry to vary the control parameter $\epsilon = 12(w/h)^2$ which is critical for observation of a vortex generation in the Stokes flow and to verify the theoretical suggestion to use 3D flow to test the 2D model. Of course, it is possible to design another microfluidic experiment with a possibility to get Re up to 3000 and channel flow past an obstacle to observe a hydrodynamic instability of a vortex generation in 3D flow and further von Karman vortex street and even pre-turbulent flow. The question is a similarity of 2D flow used in simulations and 3D flow used in the experiment in this case. If theory will show that in this case 2D flow is equivalent to 3D flow in some realistic approximation, one can test this approximation. Otherwise, there are huge number of hydrodynamic experiment of the flow past an obstacle, where theory of von Karman instability and plenty of numerical simulations at higher Re were verified.

2. It seems to me the importance of fact that the three-dimensional flow can be well-described by a 2D theoretical model is exaggerated. In the case of a planar flow and when the slab width is larger than h , this is i) a very natural fact the stems directly from conventional Navier-Stokes equation after integration over the z -axis and accounting for a parabolic velocity profile across this axis (Landau Lifshitz, Fluid Mechanics) and ii) has been already noted and addressed in detail in Ref. 4. Therefore, I suggest the authors to reconsider the discussion on this topic.

This comment is about the key basic initial point of our testing the theory. The theoretical suggestion made a possibility to present 3D Stokes flow, whether it can be described rather well by quasi-2D model, is exactly the central subject of our experiment, as we already pointed above. It is not obvious at all that the quasi-2D model obtained by an integration of the 3D channel over the channel height is very different from the 2D model used in the theory, where 2D conformal analysis is crucial. The theory made for the rectangular channel suggests the criterion at which this approach is correct. The condition $\epsilon < 120$ in the case of

the rectangular channel predicts that if the geometrical condition is satisfied, the analog of the Ohmic resistance is smaller than the viscous one, and the vortices can be generated. And this is exactly what we demonstrated in hydrodynamic flow. Of course, it is the approximation, so the agreement is semi-quantitative. And this is the reason for the quantitative discrepancy between theory and the experimental results presented in Fig.5. To conclude this comment, in Ref. 4 the similarity of the equations describing the electronic transport and Stokes equation for the fluid flow and the suggestion about the criterion for the vortex observation were made, and our experiment verified the latter.

Reviewer #1 (Remarks to the Author):

The microfluidic Stokes flow analogue of electron currents in graphene reported in this paper is elegant and intriguing. While, of course, graphene electrons and glycerol at low Reynolds are two disparate physical systems, in some regimes they obey the same coarse-grained dynamics. Thus, the main value of the present paper is in detailed measurements of a quantitatively similar system that are very hard to perform in the original one. This I think justifies publication. The authors have answered my questions and have somewhat improved the manuscript. It can definitely be further improved.

Reviewer #3 (Remarks to the Author):

The authors have provided satisfactory replies to all of my enquires and explained the significance of their investigations. Nevertheless, if the draft is accepted, I would be glad if the readers of this paper can recognize the importance of the results without the need to study the hydrodynamics of electron fluid in graphene. In particular, I still find it odd to start the paper on the viscous flow of classical fluid (1st paragraph) with the description of the consequences of frequent electron-electron collisions in graphene. In my opinion, a more direct introduction, similar to the argumentation presented in the reply file would be more appropriate.

Other than that, I don't have any further comments/recommendations and suggest the editor to make the final decision.

Reviewer #1 (Remarks to the Author):

The microfluidic Stokes flow analogue of electron currents in graphene reported in this paper is elegant and intriguing. While, of course, graphene electrons and glycerol at low Reynolds are two disparate physical systems, in some regimes they obey the same coarse-grained dynamics. Thus, the main value of the present paper is in detailed measurements of a quantitatively similar system that are very hard to perform in the original one. This I think justifies publication. The authors have answered my questions and have somewhat improved the manuscript. It can definitely be further improved.

We thank the Reviewer for recommending the manuscript for publication. We have further improved the text in the revised manuscript.

Reviewer #3 (Remarks to the Author):

The authors have provided satisfactory replies to all of my enquires and explained the significance of their investigations. Nevertheless, if the draft is accepted, I would be glad if the readers of this paper can recognize the importance of the results without the need to study the hydrodynamics of electron fluid in graphene. In particular, I still find it odd to start the paper on the viscous flow of classical fluid (1st paragraph) with the description of the consequences of frequent electron-electron collisions in graphene. In my opinion, a more direct introduction, similar to the argumentation presented in the reply file would be more appropriate.

Since our work is motivated by the viscous electron flow in graphene, therefore we first describe the motivation and the problem and then connection as well as significance to hydrodynamics of viscous Newtonian fluid flow.

We have added an additional paragraph in the Introduction to discuss viscous flow of classical fluid. It reads the following:

“The generation of vortices in a fluid flow is typically associated with high-Re flow and inertial effects. However, such inferences are likely based on the incorrect notion that low-Re flow is irrotational, which is only applicable to an ideal fluid without viscosity, where the Kelvin circulation theorem or the vorticity conservation law is valid³. Thus, it is evident that a non-potential or rotational flow bears vorticity, but to produce vortices in Stokes flow requires substantial efforts due to a strong dissipation of vorticity at $R \ll 1$ ¹⁷. Strikingly, in a wall-dominated microfluidic channel flow, vortices could be generated at low aspect ratio, w/h , of the channel despite a significant wall friction and viscous dissipation².”